# Learning Compositional Rules via Neural Program Synthesis

**Maxwell I. Nye**[*]
MIT

**Armando Solar-Lezama**
MIT

**Joshua B. Tenenbaum**
MIT

**Brenden M. Lake**
NYU
Facebook AI

## Abstract

Many aspects of human reasoning, including language, require learning rules from very little data. Humans can do this, often learning systematic rules from very few examples, and combining these rules to form compositional rule-based systems. Current neural architectures, on the other hand, often fail to generalize in a compositional manner, especially when evaluated in ways that vary systematically from training. In this work, we present a neuro-symbolic model which learns entire rule systems from a small set of examples. Instead of directly predicting outputs from inputs, we train our model to induce the explicit system of rules governing a set of previously seen examples, drawing upon techniques from the neural program synthesis literature. Our rule-synthesis approach outperforms neural meta-learning techniques in three domains: an artificial instruction-learning domain used to evaluate human learning, the SCAN challenge datasets, and learning rule-based translations of number words into integers for a wide range of human languages.

## 1 Introduction

Humans have a remarkable ability to learn compositional rules from very little data. For example, a person can learn a novel verb "to dax" from a few examples, and immediately understand what it means to "dax twice" or "dax around the room quietly." When learning language, children must learn many interrelated concepts simultaneously, including the meaning of both verbs and modifiers ("twice", "quietly", etc.), and how they combine to form complex meanings. People can also learn novel artificial languages and generalize systematically to new compositional meanings (see Figure 3). Fodor and Marcus have argued that this systematic compositionality, while critical to human language and thought, is incompatible with classic neural networks (i.e., eliminative connectionism) [1, 2, 3]. Despite advances, recent work shows that contemporary neural architectures still struggle to generalize in systematic ways when directly learning rule-like mappings between input sequences and output sequences [4, 5]. Given these findings, Marcus continues to postulate that hybrid neural-symbolic architectures (implementational connectionism) are needed to achieve genuine compositional, human-like generalization [3, 6, 7].

An important goal of AI is to build systems which possess this sort of systematic rule-learning ability, while retaining the speed and flexibility of neural inference. In this work, we present a neural-symbolic framework for learning entire rule systems from examples. As illustrated in Figure 1B, our key idea is to leverage techniques from the program synthesis community [8], and frame the problem as explicit rule-learning through fast neural proposals and rigorous symbolic checking. Instead of training a model to predict the correct output given a novel input (Figure 1A), we train our model to induce the explicit system of rules governing the behavior of all previously seen examples (Figure 1B; Grammar proposals). Once inferred, this rule system can be used to predict the behavior of any new example (Figure 1B; Symbolic application).

---

[*]Correspondence to mnye@mit.edu. Code can be found here: github.com/mtensor/rulesynthesis

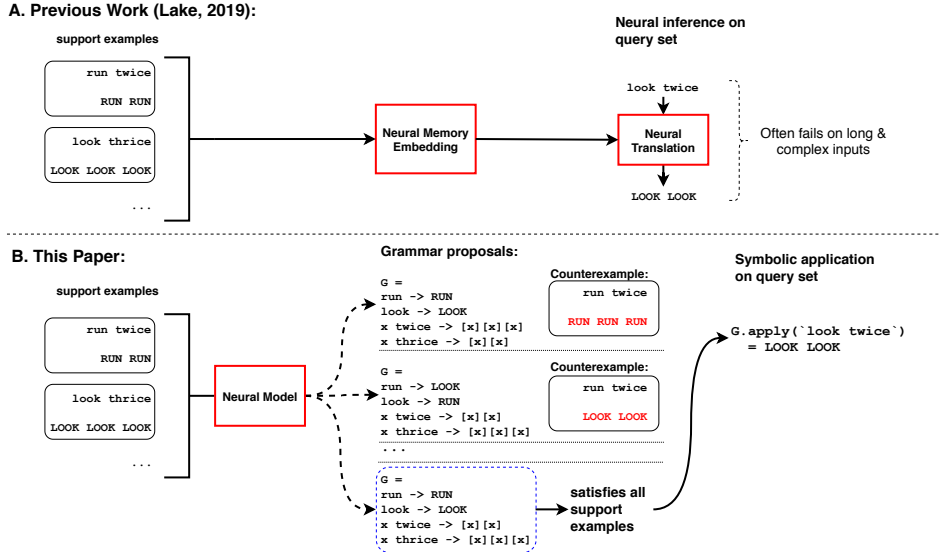

Figure 1: Illustration of our synthesis-based rule learner and comparison to previous work. A) Previous work [9]: Support examples are encoded into an external neural memory. A query output is predicted by conditioning on the query input sequence and interacting with the external memory via attention. B) Our model: Given a support set of input-output examples, our model produces a distribution over candidate grammars. We sample from this distribution, and symbolically check consistency of each sampled grammar against the support set until a grammar is found which satisfies the input-output examples in the support set. This approach allows much more effective search than selecting the maximum likelihood grammar from the network.

This explicit rule-based approach confers several advantages compared to a pure input-output approach. Instead of learning a blackbox input-output mapping, and applying it to each new query item for which we would like to predict an output (Figure 1A), we instead search for an explicit *program* which we can check against previous examples (the support set). This allows us to propose and check candidate programs, sampling programs from the neural model and only terminating search when the proposed solution is consistent with prior data.

The program synthesis framing also allows immediate and automatic generalization: once the correct rule system is learned, it can be correctly applied in novel scenarios which are a) arbitrarily complex and b) outside the distribution of previously seen examples. We draw on work in the neural program synthesis literature [10, 11] to solve complex rule-learning problems that pose difficulties for both neural and traditional symbolic methods. Our neural synthesis approach is distinctive in its ability to simultaneously and flexibly attend over a large number of input-output examples, allowing it to integrate different kinds of information from varied support examples.

Our training scheme is inspired by meta-learning. Assuming a distribution of rule systems, or a "meta-grammar," we train our model by sampling grammar-learning problems and training on these sampled problems. We can interpret this as an approximate Bayesian grammar induction, where our goal is to maximize the likelihood of a latent program which explains the data [12].

We demonstrate that, when trained on a general meta-grammar of rule-systems, our rule-synthesis method can outperform neural meta-learning techniques. Concretely, our main contributions are:

- We present a neuro-symbolic program synthesis model which can learn novel rule systems from few examples. Our model employs a symbolic program representation for compositional generalization and neural program synthesis for fast and flexible inference. This allows us to leverage search in the space of programs, for a guess-and-check approach.
- We show that our model can learn to interpret artificial languages from few examples, solving SCAN and outperforming 10 alternative models.
- Finally, we show that our model can outperform baselines in learning how to interpret number words in unseen languages from few examples.

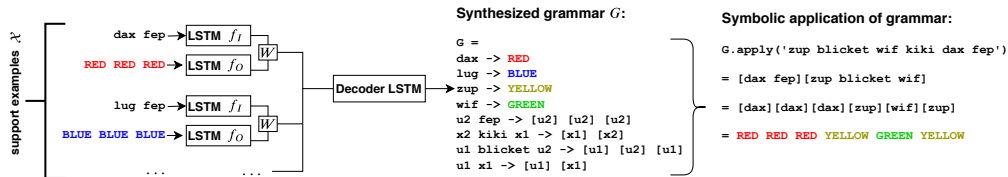

Figure 2: Illustration of our synthesis-based rule learner neural architecture and grammar application. Support examples are encoded via BiLSTMs. The decoder LSTM attends over the resulting vectors and decodes a grammar, which can be symbolically applied to held out query inputs. Middle: an example of a fully synthesized grammar which solves the task in Figure 3.

## 2 Related Work

Previous work on the SCAN challenge has employed data augmentation [14], meta-learning [9], and syntactic attention [15]. Lake [9] uses meta-learning to induce a seq-to-seq model for predicting query input-output transformations, from a small number of support examples (Figure 1A). They show significant improvements over standard seq-to-seq methods, and demonstrate that their model captures relevant human biases. Using a similar training scheme, we instead learn an explicit program which can be applied to held out query items (Figure 1B).

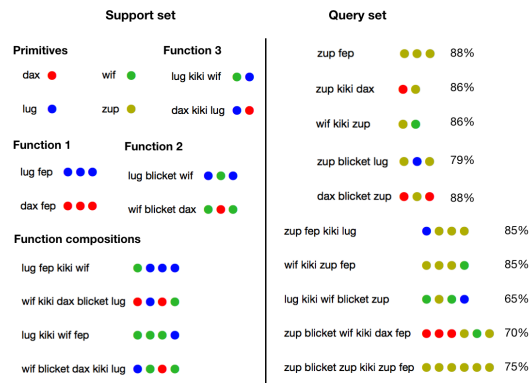

Figure 3: An example of few-shot learning of instructions. In [13], participants learned to execute instructions in a novel language of nonce words by producing sequences of colored circles. Human performance is shown next to each query instruction, as the percent correct across participants. When conditioned on the support set, our model can predict the correct output sequences on the held out query instructions by synthesizing the grammar in Figure 2.

Our approach builds on work in neural program synthesis. We are inspired by work such as RobustFill [11], enumerative approaches [16], execution guided work [17, 18, 10, 19], and hybrid models [20, 21]. A key difference in our work is the number and diversity of input-output examples provided to the system. Previous neural program synthesis systems, such as RobustFill [11], are not able to handle the large number of diverse of examples in our problems. Techniques exist for selecting examples, but they are expensive, requiring an additional outer loop of meta-learning [22], or repeating search every time a new counterexample is found (as in CEGIS [23]). Our approach uses neural attention to flexibly condition on many examples at once, without the need for an additional outer search or learning loop. This is especially relevant for our domains, given the diversity of examples, and the fact that different subsets of examples inform each rule. For a more detailed discussion of the differences between our approach and RobustFill, see Section 4.1.

There is also related work from the programming languages community, such as Sketch [23], PROSE [24], and a large class of synthesizers from the SyGuS competition [25]. However, our problems are outside the scope of domains these systems can support (integer, bit-vector and FlashFill-style string editing). Our problems are also outside the scope of functional synthesizers such as Lambda$^2$ [26] or Synquid [27]. We compare against alternative synthesis approaches in our experiments on SCAN.

## 3 Our Approach

**Overview:** Given a small support set of input-output examples, $\mathcal{X} = \{(x_i, y_i)\}_{i=1..n}$, our goal is to produce the outputs corresponding to a query set of inputs $\{q_i\}_{i=1..m}$ (see Figure 3). To do this, we build a neural program synthesis model $p_\theta(\cdot|\mathcal{X})$ which accepts the given examples and synthesizes a symbolic program $G$, which we can execute on query inputs to predict the desired query outputs, $r_i = G(q_i)$. Our symbolic program consists of an "interpretation grammar," which is a sequence of *rewrite rules*, each of which represents a transformation of token sequences. The details of the

interpretation grammar are discussed below. At test time, we employ our neural program synthesis model to drive a simple search process. This search process proposes candidate programs by sampling from the program synthesis model and symbolically checks whether candidate programs satisfy the support examples by executing them on the support inputs, i.e., checking that $G(x_i) = y_i$ for all $i = 1..n$. During each training episode, our model is given a support set $\mathcal{X}$ and is trained to infer an underlying program $G$ which explains the support and held-out query examples.

**Model:** A schematic of our architecture is shown in Figure 2. Our neural model $p_\theta(G|\mathcal{X})$ is a distribution over programs $G$ given the support set $\mathcal{X}$. Our implementation is quite simple and consists of two components: an encoder $Enc(\cdot)$, which encodes each support example $(x_i, y_i)$ into a vector $h_i$, and a decoder $Dec(\cdot)$, which decodes the program while attending to the support examples:

$$p_\theta(\cdot|\mathcal{X}) = Dec(\{h_i\}_{i=1..n}),$$
$$\text{where } \{h_i\}_{i=1..n} = Enc(\mathcal{X})$$

**Encoder:** For each support example $(x_i, y_i)$, the input sequence $x_i$ and output sequence $y_i$ are each encoded into a vector by taking the final hidden state of an input BiLSTM encoder $f_I(x_i)$ and an output BiLSTM encoder $f_O(y_i)$, respectively (Figure 2; left). These hidden states are then combined via a single feedforward layer with weights $W$ to produce one vector $h_i$ per support example:

$$h_i = ReLU(W[f_I(x_i); f_O(y_i)])$$

**Decoder:** We use an LSTM for our decoder (Figure 2; center). The decoder hidden state $u_0$ is initialized with the sum of all of the support example vectors, $u_0 = \sum_i h_i$, and the decoder produces the program token-by-token while attending to the support vectors $h_i$ via attention [28]. The decoder outputs a tokenized program, which is then parsed into an interpretation grammar.

**Interpretation Grammar:** The programs in this work are instances of an *interpretation grammar*, which is a form of term rewriting system [29]. The interpretation grammar used in this work consists of an ordered list of rules. Each rule consists of a left hand side (LHS) and a right hand side (RHS). The left hand side consists of the input words, string variables x (regexes that match entire strings), and primitive variables u (regexes that match single words). Evaluation proceeds as follows: An input sequence is checked against the rules in order of the rule priority. If the rule LHS matches the input sequence, then the sequence is replaced with the RHS. If the RHS contains bracketed variables (i.e., [x] or [u]), then the contents of these variables are evaluated recursively through the same process. In Figure 2 (right), we observe grammar application on the input sequence zup blicket wif kiki dax fep. The first matching rule is the kiki rule,[2] so its RHS is applied, producing [dax fep] [zup blicket wif], and the two bracketed strings are recursively evaluated using the fep and blicket rules, respectively.

**Search:** At test time, we sample candidate programs from our neural program synthesis model. If the new candidate program $G$ satisfies the support set —i.e., if $G(x_i) = y_i$ for all $i = 1..n$ —then search terminates and the candidate program $G$ is returned as the solution. The program $G$ is then applied to the held-out query set to produce final query predictions $r_i = G(q_i)$. During search, we maintain the best program so far, defined as the program which satisfies the largest number of support examples.[3] If the search timeout is exceeded and no program has been found which solves all of the support examples, then the best program so far is returned as the solution.

This search procedure confers major advantages compared to pure neural approaches. In a pure neural induction model (Figure 1A), given a query input and corresponding output prediction, there is no way to check consistency with the support set. Conversely, casting the problem as a search for a satisfying program allows us to explicitly check each candidate program against the support set, to ensure that it correctly maps support inputs to support outputs. The benefit of such an approach is shown in Section 4.2, where we can achieve perfect accuracy on SCAN by increasing our search budget and searching until a program is found which satisfies all of the support examples.

**Training:** We train our model in a similar manner to [9]. During each training episode, we randomly sample an interpretation grammar $G$ from a distribution over interpretation grammars, or "meta-grammar" $\mathcal{M}$. We then sample a set of input sequences consistent with the sampled interpretation grammar, and apply the interpretation grammar to each input sequence to produce the corresponding

output sequence, giving us a support set of input-output examples $\mathcal{X}_G$. We train the parameters $\theta$ of our network $p_\theta$ via supervised learning to output the grammar $G$ when conditioned on the support set of input-output examples, maximizing $\underset{(G,\mathcal{X}_G)\sim\mathcal{M}}{\mathbb{E}}\left[\log p_\theta(G|\mathcal{X}_P)\right]$ by gradient descent.

# 4 Experiments

## 4.1 MiniSCAN

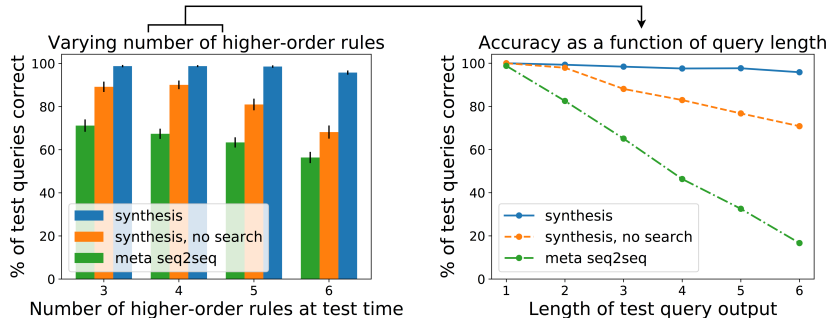

Figure 4: MiniSCAN generalization results. We train on random grammars with 3-4 primitives, 2-4 higher order rules, and 10-20 support examples. Left: At test time, we vary the number of higher-order rules. The synthesis-based approach using search achieves near-perfect accuracy for most test conditions. Right: Length generalization results. A key challenge for compositional learning is generalization across lengths. We plot accuracy as a function of query output length for the "4 higher-order rules" test condition. The accuracy of our synthesis approach does not degrade as a function of query output length, whereas the performance of baselines decreases.

Our first experimental domain is the paradigm introduced in [13], informally dubbed "MiniSCAN." The goal of this domain is to learn compositional, language-like rules from a very limited number of examples. In [13], human subjects were allowed to study the 14 example 'support instructions' in Figure 3. Participants were then tested on the 10 'query instructions' in Figure 3, to determine how well they had learned to execute instructions in this novel language. Our aim is to build a model which learns this artificial language from few examples, similar to humans. We test our model on MiniSCAN to determine how well it can induce such language-like rules systems, both when they are similar to those seen during training, as well as when they vary systematically from training data.

**Training details:** We trained our model on a series of meta-training episodes. During each episode, a grammar was sampled from the meta-grammar distribution, and our model was trained to recover this grammar given a support set of example sequences. In our experiments, the meta-grammar randomly sampled grammars with 3-4 *primitive* rules and 2-4 *higher-order* rules. Primitive rules map a word to a color (e.g. `dax -> RED`), and higher order rules encode variable transformations given by a word (e.g. `x1 kiki x2 -> [x2] [x1]`). (In a higher-order rule, the LHS can be one or two variables and a word, and the RHS can be any sequence of bracketed forms of those variables.) For each grammar, we trained with a support set of 10-20 randomly sampled examples. More details can be found in Section A.1.1 of the supplement.

**Alternate Models:** In this experiment, we compare against two closely related alternatives. The first is **meta seq2seq** [9]. This model is also trained on episodes of randomly sampled grammars. However, instead of synthesizing a grammar, meta seq2seq conditions on support examples and attempts to translate query inputs directly to query outputs in a seq-to-seq manner (Figure 1A). Meta seq2seq therefore uses a learned representation, in contrast to our symbolic program representation. The second alternate model is a lesioned version of our synthesis approach, dubbed the **no search** baseline. This model does not perform guess-and-check search, and instead returns the grammar produced by greedily decoding the most likely token at each step. This baseline allows us to determine how much of our model's performance is due to its ability to perform guess-and-check search.

**Test Details:** Our synthesis methods were tested by sampling from the network for the best grammar, or until a candidate grammar was found which was consistent with all of the support examples, using a timeout of 30 sec (on one GPU; compute details in supplemental Section A.1). We tested on 50 held-out grammars, each containing 10 query examples.

**Results:** To evaluate our rule-learning model and baselines, we test the models on a battery of evaluation schemes. In general, we observe that the synthesis methods are much more accurate than the pure neural meta seq2seq method, and only the search-based synthesis method is able to consistently predict the correct query output sequence for all test conditions. Our main results varying the number of higher order rules are shown in Figure 4, with additional results varying the number of support examples and number of primitives in the supplement (Figure A.1 To determine how well these models could generalize to grammars systematically different than those seen during training. we varied the number of higher-order functions in the test grammars (Figure 4 left). For these experiments, each support set contained 30 examples.

Both synthesis models are able to correctly translate query items with high accuracy (89% or above) when tested on held-out grammars within the training distribution (3-4 higher order rules). However, only the search-based synthesis model maintains high performance as the number of higher order rules increases beyond the training distribution, indicating that the ability to search for a consistent program plays a large role in out-of-sample generalization.

Furthermore, in instances where the synthesis-based methods have perfect accuracy because they recover exactly the generating grammar (or some equivalent grammar), they would also be able to trivially generalize to query examples of any size or complexity, as long as these examples followed the same generating grammar. On the other hand, as reported in many previous studies [30, 4, 9], approaches which attempt to neurally translate directly from inputs to outputs struggle to generate sequences much longer than those seen during training. This is a clear conceptual advantage of the synthesis approach; symbolic rules, if accurately inferred, necessarily allow correct translation in every circumstance. To investigate this property, we plot the performance of our models as a function of the query example length for the 4 higher-order rule test condition above (Figure 4 right). The performance of the baselines decays as the length of the query examples increases, whereas the search-based synthesis model experiences no such decrease in performance.

This indicates a key benefit of the program synthesis approach: When a correct program is found, it trivially generalizes correctly to arbitrary query inputs, regardless of how out-of-distribution they may be compared to the support inputs, as long as those query inputs follow the same rules as the support inputs. The model's ability to search the space of programs also plays a crucial role, as it allows the system to find a grammar which satisfies the support examples, even if it is not the most likely grammar under the neural network distribution.

We also note that our model is able to solve the task in Figure 3; we achieve a score of 98.75% on the query set, which is higher than the average score for human participants in [13]. The no search and meta seq2seq model are not able to solve the task, achieving scores of 37.5% and 25%, respectively.

**Comparison to RobustFill:** Previous neural I/O synthesis models, such as RobustFill, as well as [18, 17, 10]—designed for a small, fixed number of examples—generally use a *separate* encoder-decoder model (possibly with attention) for *each* example. Information from the separate examples is only combined through a max-pool or vector concatenation bottleneck—there is no attention *across* examples. This makes these models unsuitable for domains where it is necessary to integrate relevant information across a large number of diverse examples. To confirm this, we tested a re-implementation of the RobustFill model on MiniSCAN. Using standard hyperparameters (hidden size 512, embedding size 128, learning rate 0.001), the RobustFill model only achieves 3%, 4%, 3%, 3.5% accuracy on grammars with 3-6 higher-order rules, respectively. In contrast, our model encodes each I/O example with an example encoder, and then a single decoder model attends over these example vectors while decoding. By attending *across* examples, our approach can focus on the relevant examples at each decoding step. This is particularly important for the domains studied in this work, because there are many support examples, and only a subset are relevant at each decoding step (i.e., each rule).

## 4.2 SCAN Challenge

Our next experiments concern the SCAN dataset [4, 5]. The goal of SCAN is to test the compositional abilities of neural networks when test data varies systematically from training data. We test our model on SCAN to determine if our rule-learning approach can solve these compositional challenges.

SCAN consists of simple English commands paired with corresponding discrete actions (see Figure 5). The dataset has roughly 21,000 command-to-action examples, arranged in several test-train splits to examine different aspects of compositionality. We focus on four splits: The **simple** split randomly

```
walk                                    walk -> WALK           jump -> JUMP
WALK                                    run  -> RUN            look -> LOOK
                                        left -> LTURN          right -> RTURN
walk left twice                         turn -> EMPTY_STRING
LTURN WALK LTURN WALK                   u1 opposite u2 -> [u2] [u2] [u1]
                                        u1 around u2 ->
jump                                        [u2][u1][u2][u1][u2][u1]
JUMP                                    x2 twice -> [x2] [x2]
                                        x1 thrice -> [x1] [x1] [x1]
jump around left                        x2 after x1 -> [x1] [x2]
LTURN JUMP LTURN JUMP LTURN JUMP LTURN JUMP   x1 and x2 -> [x1] [x2]
                                        u1 u2 ->[u2] [u1]
walk right
RTURN WALK
```

Figure 5: Right: Example SCAN data. Each example consists of a synthetic language command (top) paired with a discrete action sequence (bottom). Fig. adapted from [14]. Left: Induced grammar which solves SCAN.

sorts data into the train and test sets. The **length** split places all examples with output length of up to 22 tokens into the train set, and all other examples (24 to 48 tokens long) into the test set. The **add jump** split teaches the model how to 'jump' in isolation, along with the compositional uses of other primitives, and then evaluates it on all compositional uses of jump, such as 'jump twice' or 'jump around to the right.' The **add around right** split is similar to the 'add jump' split, except the phrase 'around right' is held out from the training set. The 'add jump' and 'add around right' splits test if a model can learn to compositionally use words or phrases previously only seen in isolation.

**Training Setup:** Previous work on SCAN has used a variety of techniques [14, 9, 15]. Most related to our approach, [9] trained a model to solve related problems via meta-learning. At test time, samples from the SCAN train split were used as support items, and samples from the SCAN test split were used as query items. However, in [9], the meta-training distribution consisted of different permutations of assigning the SCAN primitive actions ('run', 'jump', 'walk', 'look') to their commands ('RUN', 'JUMP', 'WALK', 'LOOK'), while maintaining the same SCAN task structure between meta-train and meta-test. Therefore, in these experiments, the goal of the learner is to assign primitive actions to commands within a known task structure, while the higher-order rules, such as 'twice', and 'after', remain constant between meta-train and meta-test.

In contrast, we approach learning the entire SCAN grammar from few examples, by meta-training on a general and broad meta-grammar for SCAN-like rule systems, similar to our approach above in Section 4.1. Training details can be found in Section A.1.2 of the supplement.

**Testing Setup:** We test our fully trained model on each split of SCAN as if it were a new few-shot test episode with support examples and a held out query set, as above. For each SCAN split, we use the training set as test-time support elements, and input sequences from the SCAN test set are used as query elements. The SCAN training sets have thousands of examples, so it is infeasible to attend over the entire training set at test time. Therefore, at test time, we randomly sample 100 examples from the SCAN training set to use as the support set for our network. We can then run program inference, conditioned on just these 100 ex-

Table 1: Accuracy on SCAN splits.

|                     | length | simple | jump  | right |
|---------------------|--------|--------|-------|-------|
| Synth (Ours)        | **100** | **100** | **100** | **100** |
| Synth (no search)   | 0.0    | 13.3   | 3.5   | 0.0   |
| Meta Seq2Seq        | 0.04   | 0.88   | 0.51  | 0.03  |
| MCMC                | 0.02   | 0.0    | 0.01  | 0.01  |
| Sampling from prior | 0.04   | 0.03   | 0.03  | 0.01  |
| Enumeration         | 0.0    | 0.0    | 0.0   | 0.0   |
| DeepCoder           | 0.0    | 0.03   | 0.0   | 0.0   |
| GECA [14]           | –      | –      | 87    | 82    |
| Meta Seq2Seq (perm) | 16.64  | –      | 99.95 | 98.71 |
| Syntactic attention | 15.2   | –      | 78.4  | 28.9  |
| Seq2Seq [4]         | 13.8   | 99.8   | 0.08  | –     |

amples from the SCAN training set. The SCAN dataset is formed by enumerating all possible examples from the SCAN grammar up to a fixed depth; our models were trained by sampling examples from the target grammar. This causes a distributional mismatch which we rectify using heuristics to upsample shorter examples at test time, while ensuring that all rules are demonstrated. Details can be found in the supplement.

Because of the large number of training examples, we are also able to slightly modify our test-time search algorithm to increase performance: We select 100 examples as the initial support set for our network, and search for a grammar which perfectly satisfies them. If no satisfying grammar is found within a set timeout of 20 seconds, we resample another 100 support examples and retry searching for a grammar. We repeat this process until a satisfying grammar is found. This methodology, inspired

by RANSAC [31], allows us to utilize many examples in the training set without attending over thousands of examples at once.

We compare our full model with 10 alternative models, both baselines and ablations. Because the SCAN grammar lies within the support of the meta-grammar distribution, we test two probabilistic inference baselines: **MCMC** and **sampling** directly from the meta-grammar. We also test two program synthesis baselines: **enumeration** and **DeepCoder** [16]. The failure of these baselines suggests that precise recognition models are needed to search effectively in this large space; it is not enough to only predict which tokens are present in the program, as DeepCoder does. Baseline details can be found in Section A.1.2 of the supplement.

**Results:** Table 1 shows the overall performance of our model compared to baselines. Using search, our synthesis model is able to achieve perfect performance on each SCAN split. Without search, the synthesis approach cannot solve SCAN, never achieving performance greater than 15%. Likewise, meta seq2seq, using neither a program representation nor search, cannot solve SCAN when trained on a very general meta-grammar, solving less than 1% of the test set.

One advantage of our approach is that we don't need to retrain the model for each split. Once meta-training has occurred, the model can be tested on each of the splits and is able to induce a satisfying grammar for all four splits. In previous work, a separate meta-training set was used for each SCAN split (99.95% for 'jump' and 98.71% for 'right' [9]). In contrast, we meta-train once, and test on all 4 splits. Previous meta-learning approaches fail in this setting (0.51% and 0.03%).

Whereas previous approaches use the entire SCAN training set, our model requires less than 2% of the training data to solve SCAN. Supplement Table A.3 reports how many examples and how much time are required to find a grammar satisfying all support examples. Supplement Table A.4 reports running our algorithm without swapping out support sets when no perfectly satisfying grammar is found.

## 4.3 Learning Number Words

Our final experimental domain is the problem of inferring the integer meaning of a number word sequence from few examples, which provides a real-world example of compositional rule learning. See Figure 6 for an example. Our goal is to determine whether our model can learn, from few examples, the systematic rules governing number words, similar to adult human learners of a foreign language.

**Setup:** In this domain, each grammar $G$ is an ordered list of rules which defines a transformation from strings to integers (i.e, $G(\texttt{four thousand five hundred}) \to 4500$, or $G(\texttt{ciento treinta y siete}) \to 137$). We modified our interpretation grammar to allow for the simple mathematics necessary to compute integer values. Using this modified interpretation grammar, we designed a training meta-grammar by examining the number systems for three languages: English, Spanish and Chinese. More details can be found in Section A.1.3 in the supplement.

```
一 -> 1            x1 万 y1 -> [x1] * 10000 + [y1]
二 -> 2            千 y1    -> 1000 * 1 + [y1]
三 -> 3            x1 千 y1 -> [x1] * 1000 + [y1]
...               百 y1    -> 100 * 1 + [y1]
十 -> 10           x1 百 y1 -> [x1] * 100 + [y1]
百 -> 100          十 y1    -> 10 * 1 + [y1]
千 -> 1000         x1 十 y1 -> [x1] * 10 + [y1]
                  u1 x1    -> [u1] + [x1]
```

Figure 6: Induced grammar for Japanese numbers. Given the words for necessary numbers (1-10, 100, 1000, 10000), as well as 30 random examples, our system is able to recover an interpretable symbolic grammar to convert Japanese words to integers for any number up to 99,999,999.

We designed the task to mimic how it might be encountered when learning a foreign language: When presented with a core set of "primitive" words, such as the words for 1-20, 100, 1000, and a small number of examples which show how to compose these primitives (e.g., `forty five` $\to 45$ shows how to compose `forty` and `five`), an agent should be able to induce a system of rules for decoding the integer meaning of any number word. Therefore, for each train and test episode, we condition each model on a support set of primitive number words and several additional compositional examples. The goal of the model is to learn the system of rules for composing the given primitive words.

**Results:** Our results are reported in Table 2. We test our model on the three languages used to build the generative model, and test on six additional unseen languages, averaging over 5 evaluation runs for each. For many languages, our model is able to achieve perfect generalization to the held out query set. The no search baseline is able to perform comparably for several languages, however for some (Spanish, French) it is not able to generalize at all to the query set because the generated grammar is invalid and does not parse. Meta seq2seq is outperformed by the synthesis approaches.

Table 2: Accuracy on few-shot number-word learning, using a maximum timeout of 45 seconds.

|  | English | Spanish | Chinese | Japanese | Italian | Greek | Korean | French | Viet. |
|---|---|---|---|---|---|---|---|---|---|
| Synth (Ours) | **100** | **80.0** | **100** | **100** | **100** | **94.5** | **100** | **75.5** | **69.5** |
| Synth (no search) | **100** | 0.0 | **100** | **100** | **100** | 70.0 | **100** | 0.0 | **69.5** |
| Meta Seq2Seq | 68.6 | 64.4 | 63.6 | 46.1 | 73.7 | 89.0 | 45.8 | 40.0 | 36.6 |

## 5 Conclusion

We present a neuro-symbolic program synthesis model which can learn rule-based systems from a small set of diverse examples. Our approach uses neural attention to flexibly condition on many examples at once, integrating information from varied support examples. We demonstrate that our model achieves human-level performance in a few-shot artificial language-learning domain, dramatically improves upon existing benchmarks for the SCAN challenge, and successfully learns to interpret number words across several natural languages. In all three domains, the use of a program representation and explicit search provide strong out-of-sample generalization, improving upon previous neural, symbolic, and neuro-symbolic approaches. We believe that explicit rule learning is a key part of human intelligence, and is a necessary ingredient for building human-level and human-like artificial intelligence.

Future work could explore learning the meta-grammar and interpretation grammar from data, allowing our approach to be applied more broadly and with less supervision. Another important direction is to build hybrid systems that jointly learn implicit neural rules and explicit symbolic rules, with the aim of capturing the dual intuitive and deliberate characteristics of human thought [32].

## Broader Impact

Our approach involves using program synthesis to learn explicit rule systems from just a few examples. Compared to pure neural approaches, we expect that our approach has two main advantages: robustness and interpretability. Because our approach combines a neural "proposer" and a symbolic "checker", when neural inference fails, the symbolic checker can determine if the proposed program satisfies the given examples. Because the representation produced by our model is a symbolic program, it is also more interpretable than pure neural approaches; when mistakes are made, the incorrect program can be analyzed in order to understand the error. We conjecture that, if systems such as these are used in industrial or consumer settings, these interpretability and robustness features could lead to better safety and security. We hesitate to speculate on the long-term effects of such a research program, but we do not foresee certain groups of people being selectively advantaged.

## Acknowledgments and Disclosure of Funding

The authors gratefully acknowledge Kevin Ellis, Yewen Pu, Luke Hewitt, Tuan Anh Le and Eric Lu for productive conversations and helpful comments. We additionally thank Tuan Anh Le for assistance using the pyprob probabilistic programming library. M. Nye is supported by an NSF Graduate Fellowship and an MIT BCS Hilibrand Graduate Fellowship. Through B. Lake's position at NYU, this work was partially funded by NSF Award 1922658 NRT-HDR: FUTURE Foundations, Translation, and Responsibility for Data Science.

## Footnotes

[2]Note that the fep rule is not applied first because u2 is a primitive variable, so it only matches when fep is preceded by a single primitive word.

[3]Sequences which do not parse into a valid programs are simply discarded.

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
