[Supplementary Material]

# A  Supplementary Material: Learning Compositional Rules via Neural Program Synthesis

## A.1  Experimental and computational details

All models were implemented in PyTorch. All testing and training was performed on one Nvidia GTX 1080 Ti GPU. For all models, we used LSTM embedding and hidden sizes of 200, and trained using the Adam optimizer [1] with a learning rate of 1e-3. Training and testing runs used a batch size of 128. For all experiments, we report standard error below.

### A.1.1  Experimental details: MiniSCAN

**Meta-grammar**  As discussed in the main text, each grammar contained 3-4 *primitive* rules and 2-4 *higher-order* rules. Primitive rules map a word to a color (e.g. `dax -> RED`), and higher order rules encode variable transformations given by a word (e.g. `x1 kiki x2 -> [x2] [x1]`). In a higher-order rule, the left hand side can be one or two variables and a word, and the right hand side can be any sequence of bracketed forms of those variables. The last rule of every grammar is a concatenation rule: `u1 x1 -> [u1] [x1]`, which dictates how a sequence of tokens can be concatenated. Figure A.2 shows several example training grammars sampled from the meta-grammar. We trained our models for 12 hours.

**Generating input-output examples**  To generate a set of support input-output sequences $\mathcal{X}$ from a program $G$, we uniformly sample a set of input sequences from the CFG formed by the left hand side of each rule in $G$. We then apply the program $G$ to each input sequence $x_i$ to find the corresponding output sequence $y_i = G(x_i)$. This gives a set of examples $\{(x_i, y_i)\}$, which we can divide into support examples and query examples.

**Test details**  For each of our experiments, we used a sampling timeout of 30 sec, and tested on 50 held-out test grammars, each containing 10 query examples. The model samples approx. 35 prog/second, resulting in a maximum search budget of approx. 1000 candidate programs.

Figure A.1: MiniSCAN generalization results. We train on random grammars with 3-4 primitives, 2-4 higher order rules, and 10-20 support examples. At test time, we vary the number of support examples (left), primitive rules (center), and higher-order rules (right). The synthesis-based approach using search achieves near-perfect accuracy for most test conditions.

**Results**  Our results are shown in Figure A.1. We observed that, when the support set is too small, there are often not enough examples to disambiguate between several grammars which all satisfy the support set, but may not satisfy the query set. Thus, we varied the number of support examples during test time and evaluated the accuracy of each model (Figure A.1 left). We observed that, when we increased the number of support elements to 50 or more, the probability of failing any of the query elements fell to less than 1% for our model. We also we varied the number of primitives in the test grammars (Figure A.1 center), and the number of higher-order functions in the test grammars (Figure A.1 right, also Figure 4 in the main text). For these experiments, each support set contained 30 examples.

We additionally extended the higher-order rules experiment, again training on 2-4 higher-order rules, but testing on grammars with 7-13 higher-order rules. See Table A.1 for results. These results demonstrate generalization and relatively graceful degradation on test grammars with up to 3x the number of rules compared to those seen during training.

Table A.1: Accuracy on extended higher-order rules MiniSCAN experiment, with standard error.

| Higher-order rules: | 7 | 8 | 9 | 10 | 11 | 12 | 13 |
|---|---|---|---|---|---|---|---|
| Synth (Ours) | **96.0** (1.3) | **93.6** (1.4) | **92.0** (1.7) | **90.5** (2.1) | **83.5** (3.4) | **78.5** (3.6) | **77.5** (3.1) |
| Synth (no search) | 59.5 (5.7) | 62.0 (2.8) | 62.5 (4.4) | 56.0 (4.3) | 59.5 (4.7) | 48.5 (3.8) | 52.5 (4.5) |
| Meta Seq2Seq | 58.5 (3.6) | 59.8 (2.3) | 69.0 (4.5) | 62.5 (3.9) | 56.5 (4.2) | 55.5 (3.7) | 53.0 (4.3) |

### A.1.2  Experimental details: SCAN

**Meta-grammar**   The meta-grammar used to train networks for SCAN is based on the meta-grammar used in the MiniSCAN experiments above. Each grammar has between 4 and 9 primitives and 3 and 7 higher order rules, with random assignment of words to meanings. Examples of random grammars are shown below. Models are trained on 30-50 support examples, and we train for 48 hours, viewing approximately 9 million grammars.

This meta-grammar has two additional differences from the MiniSCAN meta-grammar, allowing it to produce grammars which solve SCAN:

1. Primitives can rewrite to empty tokens, e.g., `turn -> EMPTY_STRING`.

2. The last rule for each grammar can either be the standard concatenation rule above, or, with 50% probability, a different concatenation rule: `u1 u2 -> [u2] [u1]`, which acts only on two adjacent single primitives. This is to ensure that the SCAN grammar, which does not support general string concatenation, is within the support of the training meta-grammar, while maintaining compatibility with MiniSCAN grammars.

Example training grammars sampled from the meta-grammar are shown in Figure A.3. At training time, we use the same process as for MiniSCAN to sample input-output examples for the support and query set.

**Selecting support examples at test time**   The distribution of input-output example sequences in each SCAN split is very different than the training distribution. Therefore, selecting a random subset of 100 examples uniformly from the SCAN training set would lead to a support set very different from support sets seen during training. We found that two methods of selecting support examples from each SCAN training set allowed us to achieve good performance:

1. To ensure that support sets during testing matched the distribution of support sets during training, we selected our test-time support examples to match the empirical distribution of input sequence lengths seen at training time. We used rejection sampling to ensure consistent sequence lengths at train and test time.

2. We found that results were improved when words associated with longer sequences were seen in more examples in the test-time support set. Therefore, we upweighted the probability of seeing the words 'opposite' and 'around' in the support set.

The implementation details of support example selection can be found in `generate_episode.py`.

**Baselines**   Our probabilistic baselines were implemented in the `pyprob` probabilistic programming language [2]. For both baselines, we allow a maximum timeout of 180 seconds. Both MCMC and sampling evaluate more candidate programs than our baseline, achieving about 60 programs/sec, compared to the synthesis model, which evaluates about 35 programs/sec.

For our enumeration and DeepCoder baselines, we used an optimization, inspired by CEGIS [3], to increase enumeration speed. When checking candidate grammars against the support set examples, we randomly selected 4 examples from the support set, and only checked the grammar against those 4 examples. We only checked the grammar against the other support set examples if any of the original 4 examples were satisfied. Using this optimization, our enumeration and DeepCoder baselines enumerated approximately 1000 programs/sec. For the enumerative baselines, we also allow a maximum timeout of 180 seconds.

Table A.2: Accuracy on SCAN splits with standard error.

|  | length | simple | jump | right |
|---|---|---|---|---|
| Synth (Ours) | **100** | **100** | **100** | **100** |
| Synth (no search) | 0.0 | 13.3 (3.3) | 3.5 (0.7) | 0.0 |
| Meta Seq2Seq | 0.04 (0.02) | 0.88 (0.13) | 0.51 (0.06) | 0.03 (0.03) |
| MCMC | 0.02 (0.01) | 0.0 | 0.01 (0.01) | 0.01 (0.01) |
| Sample from prior | 0.04 (0.02) | 0.03 (0.03) | 0.03 (0.02) | 0.01 (0.01) |
| Enumeration | 0.0 | 0.0 | 0.0 | 0.0 |
| DeepCoder | 0.0 | 0.03 (0.02) | 0.0 | 0.0 |

Table A.3: Required search budget for our synthesis model on SCAN, with standard error.

|  | length | simple | jump | right |
|---|---|---|---|---|
| Search time (sec) | 39.1 (11.9) | 33.7 (10.0) | 74.6 (48.5) | 36.1 (13.4) |
| Number of prog. seen | 1516 (547) | 1296 (358.2) | 2993 (1990.1) | 1466 (541) |
| Number of ex. used | 149.4 (28.9) | 144.8 (24.7) | 209.2 (91.3) | 143.8 (28.6) |
| Frac of ex. used | 0.88% | 0.86% | 1.6% | 0.94% |

**Results** Table A.2 and shows the numerical results for the SCAN experiments in the main paper, reported with standard error. Table A.3 reports how many examples and how much time are required to find a grammar satisfying all support examples. Table A.4 shows the fixed example budget results, averaged over 20 evaluation runs. Under this test condition, we achieve perfect performance on the length and simple splits within 180 seconds, and nearly perfect performance on the add around right split (98.4%). The add jump split is more difficult; we achieve 43.3% ($\pm$10%) accuracy.

We also ran an experiment to further investigate the distributional mismatch between training and testing examples. As discussed in the main text, the SCAN splits were formed by enumerating examples from the SCAN grammar up to a fixed depth, whereas our models were trained by sampling examples from the target training grammar. At test time, we used example-selection heuristics to rectify this distributional mismatch. In this additional experiment, we test whether our model can synthesize the SCAN grammar without these heuristics, provided the distributional mismatch is controlled for. We constructed a new SCAN corpus by re-generating data by *sampling* examples from the SCAN grammar instead of enumerating, and randomly assigning sampled examples to the train or test set.[1] We observe that our model is able to solve this corpus without the example-selection heuristics described above. Following the methodology in Table A.3, we find that, to achieve perfect accuracy on this "sampled" SCAN corpus, we require a search budget of 69.2 seconds ($\pm$ 16.1), 2146 programs ($\pm$ 515), and 255.8 examples ($\pm$ 39.0).

### A.1.3 Experimental details: Number Words

**Meta-grammar** We designed a meta-grammar for the number domain, relying on knowledge of English, Spanish, and Chinese. The meta-grammar includes features common to these three languages, including regular and irregular words for powers of 10 and their multiples, exception words, and features such as zeros or conjunctive words. We assume a base 10 number system, where powers of 10 can have "regular" words (e.g., "one hundred", "two hundred", "three hundred" ) or "irregular" words ("ten", "twenty", "thirty"). Additional features include exceptions to regularity,

Table A.4: Accuracy on SCAN splits, using a fixed budget of 100 examples.

| Model | length | simple | jump | right |
|---|---|---|---|---|
| Synth (180 s) | **100** | **100** | **43.3** (10.0) | **98.4** (1.6) |
| Synth (120 s) | **100** | 98.4 (1.6) | **53.9** (10.3) | 94.2 (2.9) |
| Synth (60 s) | 92.2 (3.8) | 97.5 (1.3) | **44.3** (9.6) | 80.75 (6.8) |
| Synth (30 s) | 85.6 (4.6) | 95.6 (2.3) | 24.2 (8.6) | 60.0 (8.7) |

Table A.5: Accuracy on few-shot number-word learning, using a maximum timeout of 45 seconds. Results shown with standard error over 5 evaluation runs.

| Model | English | Spanish | Chinese | Japanese | Italian | Greek | Korean | French | Viet |
|---|---|---|---|---|---|---|---|---|---|
| Synth (Ours) | **100** | **80.0** (17.9) | **100** | **100** | **100** | **94.5** (4.9) | **100** | **75.5** (2.4) | **69.5** (2.3) |
| Synth (no search) | **100** | 0.0 | **100** | **100** | **100** | 70.0 (10.2) | **100** | 0.0 | **69.5** (2.3) |
| Meta Seq2Seq | 68.6 (10.0) | 64.4 (3.2) | 63.6 (4.0) | 46.1 (3.5) | 73.7 (3.2) | 89.0 (2.5) | 45.8 (3.7) | 40.0 (5.3) | 36.6 (6.2) |

conjunctive words (e.g., "y" in Spanish), and words for zero. The full model can be found in `pyro_num_distribution.py`, and example training grammars are shown in Figure A.4.

**Training**  We trained our model on programs sampled from the constructed meta-grammar. For each training program, we sampled 60-100 string-integer pairs to use as support examples, and sampled 10 more pairs as held-out query set. We train and test on numbers up to 99,999,999. We trained all models for 12 hours.

**Test Setup**  To test our trained model on real languages, we used the PHP international number conversion tool to gather data for several number systems. On the input side, the neural model is trained on a large set of input tokens labeled by ID; at test time, we arbitrarily assign each word in the test language to a specific token ID. Character-level variation, such as elision, omission of final letters, and tone shifts were ignored. For integer outputs, we tokenized integers by digit. For testing, we conditioned on a core set of primitive examples, plus 30 additional compositional examples. At test time, we increased the preference for longer compositional examples compared to the training time distribution, in order to test generalization.

**Generating input-output examples**  For each grammar, example pairs $(x_i, y_i)$ come in two categories: a core set of "necessary" primitive words, and a set of compositional examples.

1. Necessary words: The core set of "necessary words" are analogous to the primitives for the MiniSCAN and SCAN domains. This set comprises examples with only one token as well as examples for powers of 10. For both training and testing, we produce an example for every necessary word in the language. For the synthesis models, we automatically convert the core primitive examples into rules.

2. Compositional examples: At test time, to provide random compositional examples for each language, we sample numbers from a distribution over integers and convert them to words using the `NumberFormatter` class (see `convertNum.php`). To ensure a similar process during training time, to produce compositional example pairs $(x_i, y_i)$ for a training grammar $G$, we sample numbers $y_i$ from a distribution over integers. We then construct the inverse grammar $G^{-1}$, which transforms integers to words, and use this to find the input sequence examples $x_i = G^{-1}(y_i)$. At test time, the compositional example distribution is slightly modified to encourage longer compositional examples. The sampling distribution can be found in `test_langs.py`. At training time, we produce between 60 and 100 compositional examples for the support set, and 10 for the held out query set. At test time, we produce 30 compositional examples for the support set and 30-70 examples for the held out query set.

**Results**  Table A.5 shows the results in the number word domain with standard error, averaged over 5 evaluation runs for each language.

```
G =
mup -> BLACK
kleek -> WHITE
wif -> PINK
u2 dax u1 -> [u1] [u1] [u2]
u1 lug -> [u1]
x1 gazzer -> [x1]
u2 dox x1 -> [x1] [u2]
u1 x1 -> [u1] [x1]

G =
tufa -> PINK
zup -> RED
gazzer -> YELLOW
kleek -> PURPLE
u2 mup x2 -> [u2] [x2]
x2 dax -> [x2]
u2 lug x2 -> [u2] [x2]
u1 dox -> [u1] [u1] [u1]
u1 x1 -> [u1] [x1]

G =
gazzer -> PURPLE
wif -> BLACK
lug -> GREEN
x2 kiki -> [x2] [x2]
x1 dax x2 -> [x2] [x1]
x1 mup x2 -> [x2] [x1] [x2] [x1] [x1]
u1 x1 -> [u1] [x1]
```

Figure A.2: Samples from the training meta-grammar for MiniSCAN.

```
G =
turn -> GREEN
left -> BLUE
right -> WALK
thrice -> RUN
blicket -> RED
u2 and x1 -> [x1] [x1] [x1] [u2] [u2] [u2] [x1]
u1 after x2 -> [u1] [u1] [x2] [x2] [u1] [x2] [x2]
u2 opposite -> [u2] [u2]
u1 lug x2 -> [u1] [x2]
u1 x1 -> [u1] [x1]

G =
and -> JUMP
kiki -> LTURN
blicket -> BLUE
walk -> LOOK
thrice -> RED
run -> GREEN
dax -> RUN
after -> RTURN
x2 twice u1 -> [u1] [x2] [x2] [x2] [x2]
u2 right x1 -> [x1] [u2] [u2]
u1 look x2 -> [u1] [x2] [x2]
u1 jump -> [u1] [u1]
u2 turn u1 -> [u2] [u1]
u1 lug -> [u1] [u1]
x2 left u1 -> [x2] [u1]
u1 x1 -> [u1] [x1]

G =
twice -> WALK
jump -> RTURN
turn -> JUMP
walk ->
blicket -> GREEN
kiki -> RUN
right -> RED
run -> BLUE
x2 left -> [x2] [x2] [x2] [x2] [x2]
x1 dax u1 -> [u1] [x1] [u1]
u1 thrice x2 -> [u1] [x2] [x2] [u1] [u1]
x1 look u2 -> [x1] [x1] [u2] [x1]
x2 around -> [x2]
u1 u2 -> [u2] [u1]

G =
twice -> WALK
jump -> RTURN
turn -> JUMP
walk ->
blicket -> GREEN
kiki -> RUN
right -> RED
run -> BLUE
x2 left -> [x2] [x2] [x2] [x2] [x2]
x1 dax u1 -> [u1] [x1] [u1]
u1 thrice x2 -> [u1] [x2] [x2] [u1] [u1]
x1 look u2 -> [x1] [x1] [u2] [x1]
x2 around -> [x2]
u1 u2 -> [u2] [u1]
```

Figure A.3: Samples from the training meta-grammar for SCAN.

```
G =
token14 -> 1
token16 -> 2
token50 -> 3
token31 -> 4
token49 -> 5
token28 -> 6
token17 -> 7
token03 -> 8
token06 -> 9
token14 token10 -> 10
token13 -> 100
token14 token36 -> 1000
token01 -> 1000000
token08 y1 -> 1000000* 1 + [y1]
token05 token01 y1 -> 1000000* 9 + [y1]
x1 token01 y1 -> [x1]*1000000 + [y1]
x1 token36 y1 -> [x1]*1000 + [y1]
token32 y1 -> 100* 1 + [y1]
x1 token13 y1 -> [x1]*100 + [y1]
x1 token10 y1 -> [x1]*10 + [y1]
u1 token09 x1 -> [u1] + [x1]
u1 x1 -> [u1] + [x1]

G =
token20 -> 1
token22 -> 2
token37 -> 3
token14 -> 4
token01 -> 5
token13 -> 6
token48 -> 7
token05 -> 8
token16 -> 9
token47 -> 10
token07 -> 20
token08 -> 30
token35 -> 40
token02 -> 50
token40 -> 60
token31 -> 70
token43 -> 80
token29 -> 90
token20 token38 -> 100
token20 token18 -> 1000
token20 token33 -> 10000
token28 token33 y1 -> 10000* 7 + [y1]
x1 token33 y1 -> [x1]*10000 + [y1]
x1 token18 y1 -> [x1]*1000 + [y1]
x1 token38 y1 -> [x1]*100 + [y1]
u1 x1 -> [u1] + [x1]
```

Figure A.4: Samples from the training meta-grammar for number word learning. Note that the model is trained on a large set of generic input tokens labeled by ID. At test time, we arbitrarily assign each word in the test language to a specific token ID.

## Footnotes

[1]This corpus is therefore analgous to the "Simple" split.