[Reviews · NeurIPS 2020]

Review 1

Summary and Contributions: The authors train a model on data to improve its systematic generalisation capability (i.e. out-of-distribution generalisation) by first inferring a symbolic rule system (a grammar) which is then used in a classic way to solve the problem. In this work, all experiments centre around the synthetic toy dataset SCAN (or very similar) which has been an interesting benchmark in recent years. SCAN can be solved by RNNs if they are trained on enough examples, but most previous methods fail when the training and test data contain systematic differences. The contribution of this paper is an extension of a previous line of work which makes use of attention mechanisms and clever "meta-learning" objectives. Through stronger symbolic constraints (and additional information about the problem) the authors demonstrate almost perfect performance on the toy tasks. In summary, I believe that the authors proceed as follows: s1. Define a class/family/set M of grammars (also called a meta-grammar in the paper) s2. During a training episode, sample first a grammar G from that set. s3. Use G to generate your training data for this episode (including a supporting set with examples) s4. Train a neural network to predict G. The model is trained for 48h on roughly 9 million different grammars. ---- testing ---- s5. To test, a supporting set is sampled from the training data (100 samples) based on a) similar distribution in example lengths and b) they manually make special long sequence words more likely to be picked for the supporting set. (!) s6. From the supporting set, a grammar G is generated using the trained neural network. s7. Using various supporting sets, various grammars can be sampled. s8. The grammars are used to "parse" the query samples (with or without search). s9. The final performance is given by the best grammar or the first grammar which is consistent with all samples of the support set.

Strengths: The paper is well written and builds on recent previous work. The proposed method achieves a generalisation to longer sequence lengths which was a critical weakness of previous work. Their experimental evaluation seems sound and validates their method. The motivation is highly relevant for the connectionist NeurIPS community. The paper provides a detailed appendix with useful details and examples, as well as, working pytorch code of acceptable quality.

Weaknesses: w1.) The paper claims that the model "learn[s] entire rule systems from a small set of examples". I'm not convinced that this is the case in this work and neither in the previous work which this one extends (i.e. [9]). Both methods heavily rely on the supporting set and the specific neural attention architecture of the encoder and decoder which allow for the replacement of individual tokens. This allows the model to exploit a certain pattern in the support set e.g. "a b c -> a c a" by replacing the "a" and "b" on-the-fly and execute the abstract rule given by the supporting set. If the abstract rule is not present in the supporting set, the model is not able to use it. As such, the supporting set acts a lot like a library. I believe this interpretation is supported by three points: w1.1) the architecture is explicitly designed for symbol replacement through two attention mechanisms w1.2) the performance of the model breaks down if the supporting set doesn't cover all abstract rules which require the manual tweaks (see 5a,5b) in the case of systematically different test sets. w1.3) the performance is only reported w.r.t the best grammar but many grammars are sampled which seems to indicate that some grammars perform badly which is possibly due to the lack of a "complete library". I'm not aware of any argument by the authors why the model should be considered of actually learning the rules. w2.) The model definition is lacking. From the code, in model.py the encoder seems to be BatchedRuleSynthEncoderRNN and the decoder BatchedDoubleAttnDecoderRNN but section 3 is not really giving justice to the complexity of those models (see metanet_attn.py which goes cleary beyond a simple lstm with attention). In this sense, the model appears to be more complex and problem-specific than presented. w3.) This work, just like in [9], the authors incorporate additional information about the model through the strong supervision given by the meta-grammar M. The authors have a deep understanding of the structure of the data and, essentially, use an informed data augmentation and model bias to enforce certain generalisation capability. For natural data, such a strong supervision signal is not given which seems to be a severe limitation. It is unclear to me to which extend the compared methods make use of or have access to this data augmentation bias.

Correctness: I have not found any incorrect statements.

Clarity: The paper is clear and explains the experiments thoroughly. The approach is lacking to the extend that I'm doubtful that it would be reproducable but the authors included code which is certainly readable.

Relation to Prior Work: The relation to prior work is made clear.

Reproducibility: Yes

Additional Feedback: two technical questions: q1.) Why is there a drop in accuracy for Spanish and French? What makes the samples for those languages different? q2.) How is it ensured that the grammar is properly formatted? The grammar seems to be generated on a token-by-token basis which could produce illegal grammars. How is this prevented? suggestions: c1.) "program" and "grammar" seems at places to be used interchangably. The distinction, if any, is not clear to me and the text might be improved at such places. c2.) the text in figure 1 and 3 is too small final comments: Despite the more critical tone of this review, I believe this work to be valuable and worthy of publication at NeurIPS. The issues addressed are important and I consider the present method sufficiently novel and interesting. That said, I'd welcome it if the authors would highlight better the limitations of their method. ---- post rebuttal ---- After the reviewer discussion and the author rebuttal I have decided to leave my score at a 7. I fully approve the arguments raised by the other reviewers and I endorse the fact that the authors agreed to address the heuristics and the strong supervision in the main text.


Review 2

Summary and Contributions: This paper is about learning compositional rules from small amounts of data with a program synthesis approach, by learning a model that takes the data (examples of how to interpret compositional commands) and generates a grammar that explains the commands. By placing a very strong inductive bias on the space of outputs for the model, the paper shows that the proposed method can perform very well compared to prior approaches on the SCAN dataset. The paper also evaluates on a simplified version of SCAN and a dataset of number words and shows good performance when compared to baselines. While the paper uses existing neural program synthesis methods, it convincingly shows that they can work very well on problems like the SCAN dataset.

Strengths: The paper demonstrates very strong results on the SCAN benchmark, exhibiting 100% accuracy on all train/test splits which greatly exceeds all prior methods. It shows quite convincingly that neural program synthesis methods can infer grammars like the one used in SCAN from a relatively small amount of examples. The evaluation also includes other program synthesis-related baselines (such as DeepCoder and enumerative search) which shows that a more powerful approach like used in the paper is important. It is also nice to see a demonstration on a real-world number dataset, to show that the approach works on non-toy data.

Weaknesses: The approach clearly works for SCAN, but I wonder if it ends up being too specialized for it. The method clearly fails if the space for the grammar (the programming language that the synthesizer outputs) is too limited (c.f. the additions to the meta-grammar compared to Mini-SCAN in lines 41-46 of the supplementary material). Conversely, the method might not work as well if the space of the grammar becomes too big, e.g. if we add more types of variables, because it becomes more likely that there will be grammars that are correct on the support set but fail to generalize to the rest, and the synthesizer might not be able to learn the space of grammars as well. There is also no held-out component to SCAN, so it's not possible to prevent the developers of an approach like in the paper from iterating on the details of the grammar space, program synthesis model, etc. until the accuracy reaches a desirable level. Considering that SCAN is a synthetic dataset, there also isn't too much to gain from engineering to do better specifically on it. Now that this paper has shown how to solve SCAN, maybe the right answer for the community is to move towards harder and more realistic datasets. In particular, real data is likely to have noise, so the search-based filtering wouldn't work any more. Language also doesn't exhibit perfect compositionally (for example, with idioms, or with the "irregular" number words mentioned on line 85 of the supplementary material). It will be interesting to see how methods like the one in this paper can adapt to more challenging scenarios like this.

Correctness: From the description in the paper, the methodology seems sound.

Clarity: Overall the paper is clear. Lines 142-144: I didn't understand why a pure neural model wouldn't allow for checking consistency with the support set. Can't you just run the model on the support set and see whether it gives the correct results? Is the issue that there's no good way to use the results of this check, because e.g. the model might have just overfit to the support set, or when the model exhibits a failure on the support set, there's no good way to fix the model or get an alternative model to use instead? How are the tokens encoded for the LSTMs? If they are embeddings looked up by the string of the token, then do we have to take care to ensure that the terminals used in the meta-grammar are the same as used in the test data?

Relation to Prior Work: I am not intimately familiar with the prior work relating to SCAN, but I didn't notice any issues with the discussion of the prior work.

Reproducibility: Yes

Additional Feedback: Thank you for providing the code!


Review 3

Summary and Contributions: This paper proposes to synthesize compositional rules for sequence-to-sequence tasks. Instead of directly predicting output sequences from input sequences, their model generates the grammar, which could be applied to the input sequences and produce the corresponding outputs. Two key benefits of this formulation are: (1) once the correct grammar is predicted, it can fully generalize to any sample consistent with the training data; and (2) the grammar can be verified to see whether it is consistent with the training samples, so the model does not necessarily need to output the top-1 prediction based on the decoding probability. At a high level, their framework is very similar to existing work on neural program synthesis from input-output examples, except that they synthesize grammar rules within a meta-grammar space, while program synthesis work generates program in some programming languages. For model training, they design a meta-grammar space that covers the target grammar for evaluation, and they train the model by first sampling a grammar from the meta-grammar space, then generating some corresponding input-output examples as the model input. This training paradigm is also commonly used for neural program synthesis. They evaluate their Synth approach on 3 domains: (1) MiniSCAN for few-shot learning; (2) SCAN benchmark that emphasizes compositional generalization; and (3) natural language words to number translation benchmark that they construct themselves. In general, their approach outperforms other meta learning and search baselines for these tasks. In particular, when allowing the model to sample >1000 grammars and select the one that satisfies the input data, their Synth approach achieves 100% accuracy on all SCAN splits, which emphasize compositional generalization.

Strengths: Some existing work shows that sequence-to-sequence models themselves do not generalize compositionally. For example, after how to translate "jump" and "look twice", the model does not necessarily know how to translate "jump twice". Although some recent work proposes new model architectures and training algorithms to improve the compositional generalization, they still may not achieve 100% accuracy in some scenarios. I believe learning to explicitly generate symbolic rules is the right direction towards compositional generalization.

Weaknesses: To me, the approach is suitable for rule synthesis in the few-shot learning setting, i.e., MiniSCAN and word to number translation tasks. However, I don't think their approach is a good solution to SCAN tasks. Specifically, their model is very similar to the standard approach for neural program synthesis from input-output examples, which typically assumes that the number of input-output examples is small, e.g., no more than 10. Although the Synth model takes up to 100 samples simultaneously as the model input, I don't think it is qualitatively different from prior work, and it is merely a problem-dependent choice. By design, this type of models cannot take many input-output pairs into consideration at the same time, i.e., >10,000 in total on SCAN. Therefore, each time they sample 100 examples to synthesize the grammar. Though they achieve good empirical results when sampling >1000 grammars during inference, note that it requires some problem-specific assumptions, e.g., they need some heurstics to sample input-output sequence pairs for rule synthesis (see Appendix A.1.2). The results of Synth (no search) somehow further suggest that the model does not really learn the grammar rules correctly. Note that in the literature of neural program synthesis, the performance of greedy decoding is mostly reasonable though could be sub-optimal. Meanwhile, for all the three benchmarks, I feel that the meta-grammar spaces are pretty problem-specific. On MiniSCAN, they show results of training on grammars with 2-4 higher-order rules, and then testing on 5-6 higher-order rules. How do the results look like when testing on grammars with more higher-order rules, e.g., more than 10? Additional comments: For natural language to number translation, I feel that the number words in Chinese and Japanese are identical, at least from the examples. Figure 5: in the caption, "Left" and "Right" should be swapped.

Correctness: The approach is technically sound.

Clarity: In general the paper is well-written.

Relation to Prior Work: The paper provides a good discussion of related work. My main question is about this sequence in the Introduction: "Our neural synthesis approach is distinctive in its ability to simultaneously and flexibly attend over a large number of input-output examples, allowing it to integrate different kinds of information from varied support examples." However, I don't see any significant difference of their model design compared to existing neural program synthesis work.

Reproducibility: Yes

Additional Feedback:

[Author Response · NeurIPS 2020]

We would like to thank all three reviewers for their thoughtful comments. We are pleased with the generally positive reception of our work, and we will make sure to incorporate all of the helpful feedback into the camera-ready version.

**Relation to prior synthesis approaches.** R3 saw our approach as "very similar to the standard approach for neural program synthesis from input-output examples, which typically assumes that the number of input-output examples is small, e.g., no more than 10." We believe our model actually differs significantly from previous approaches in this regard. Previous neural I/O synthesis models, such as RobustFill, as well as [17], [18], [10]—designed for a small, fixed number of examples—generally use a *separate* encoder-decoder model (possibly with attention) for *each* example. Information from the separate examples is only combined through a max-pool or vector concatenation bottleneck—there is no attention *across* examples. This makes these models unsuitable for domains where it is necessary to integrate relevant information across a large number of diverse examples. To confirm this, we implemented RobustFill on our MiniSCAN domain, which only achieves 3%, 4%, 3%, 3.5% accuracy on 3-6 higher-order rules, respectively. In contrast, our model encodes each I/O example with an example encoder (ln 115), and then a single decoder model (ln 119) attends over these example vectors while decoding. By attending *across* examples, our approach can focus on the relevant examples at each decoding step. This is particularly important for our domains because there are many examples, and only a subset are relevant for each decoding step (i.e., each rule). We agree that we did not highlight this distinction enough in our submission; we thank R3 for pointing this out, and will incorporate this discussion and new RobustFill results into our paper. R1 notes that our submitted code seems to implement a more complicated network than is described in our submission. While our code is able to perform a "double attention" mechanism, this work does not use these features of the code, and the architecture is as described in the paper. We thank R1 and apologize for this confusion.

**Suitability of our approach to SCAN.** According to R2, our paper "shows quite convincingly that neural program synthesis methods can infer grammars like the one used in SCAN from a relatively small amount of examples." On the other hand, R3 questions if our approach is suitable for SCAN because it only considers up to 100 examples at a time, whereas the SCAN training sets contain >10,000 examples. We agree with R2; we see this as an advantage of our system rather than a weakness. While previous approaches require training on tens of thousands of examples, our results demonstrate that using our method, 100 examples are enough to achieve perfect performance on 4 SCAN splits.

**Support set selection.** R1 and R3 also note that we use heuristics to select support set examples for SCAN. As discussed in the supplement (ln 50-61), our heuristics are guided by general principles: we seek to match the distribution of examples in the train and test sets. The SCAN dataset is formed by *enumerating* all possible examples from the SCAN grammar up to a fixed depth; our models were trained by *sampling* examples from the target grammar. This causes a distributional mismatch which we rectify by upsampling shorter examples at test time, while ensuring that all rules are demonstrated. We ran an additional experiment, re-generating the SCAN data by sampling instead of enumerating. If we condition our model on support examples *sampled* from the underlying SCAN grammar, we are also able to solve SCAN from 100 examples. Our revision will report this experiment and move the discussion on the heuristics to the main text.

**Search.** Our approach utilizes test-time search, which R3 also suggests is a disadvantage: "The results of [the no search baseline] somehow further suggest that the model does not really learn the grammar rules correctly." We respectfully disagree. We see the entire neuro-symbolic system—rather than just the neural component—as the model; in particular, the search procedure is an asset of our method. Indeed, our results demonstrate how, because we use test-time search, our neural network does not need to be perfectly trained for our overall model to yield excellent prediction accuracy. We contrast this with neural-only approaches (as well as the no-search baseline) which require very careful training in order to achieve perfect results. In that sense, our approach offers more robustness than a neural-only model would allow.

**Meta-grammar.** The reviewers note that our model uses strong supervision in the form of a meta-grammar. We have shown how training on examples from a family of grammars can lead to very high accuracy. Although used in this work, this amount of strong supervision may not be needed. We see our work as an important step in a larger line of work, which can extend our approach to: incorporate grammars which support noise (R2), incorporate RL to replace much of the strong supervision, and learn the execution model. In a sense, we agree with R2: "Now that this paper has shown how to solve SCAN, maybe the right answer for the community is to move towards harder and more realistic datasets." R3 suggests extending our MiniSCAN experiment to test on grammars with many more rules than the 2-4 in the training meta-grammar. We report % accuracy here, for testing on 7-13 higher-order rules, respectively: synth: 96.0, 93.6, 92.0, 90.5, 83.5, 78.5, 77.5; no search: 59.0, 62.0, 62.5, 56.0, 59.5, 48.5, 52.5; meta Seq2Seq: 58.5, 59.8, 69.0, 62.5, 56.5, 55.5, 53.5. This demonstrates both generalization and graceful degradation on grammars with 3x the number of rules vs training.

**Additional questions.** R3 asks if the Chinese and Japanese number systems are the same; they are not. For example, they differ when the leading digit is 1, or when 0 is a middle digit. R2 asks how tokens are encoded. See supp. ln 93-97 for details on how we can tokenize in a symbol-agnostic manner. R1 asks how we ensured that grammars are properly formatted. If the model predicts a sequence which does not parse into a valid grammar, it is simply discarded and another is sampled. R1 asks about errors in the number word experiments. French, for example, contains irregularities such as 80 → quatre-vingts ("four twenty"), which our grammar did not support. We will discuss this in camera-ready.

[Meta-Review · NeurIPS 2020]

Learning compositional rules is an important research direction. The proposed method achieves 100% accuracy on different train/test splits of SCAN. My main concern on this work is it seems to be too specific for SCAN, as pointed out by the reviewers.